# communications
# engineering

# Ladder-shaped microfluidic system for rapid antibiotic susceptibility testing

Ann V. Nguyen[1], Mohammad Yaghoobi[1], Morteza Azizi[1], Maryam Davaritouchaee[1], Kenneth W. Simpson[2] & Alireza Abbaspourrad [1✉]

Rapid identification of antibiotic-resistant bacteria will play a key role in solving the global antibiotic crisis by providing a route to targeted antibiotic administration. However, current bacterial infection diagnoses take up to 3 days which can lead to antibiotic treatment that is less effective. Here we report a microfluidic system with a ladder shaped design allowing us to generate a twofold serial dilution of antibiotics comparable to current national and international standards. Our consolidated design, with minimal handling steps cuts down the testing time for antibiotic susceptibility from 16–20 h to 4–5 h. Our feasibility testing results are consistent with the commercial antibiotic susceptibility testing (AST) results, showing a 91.75% rate of agreement for Gram-negative and Gram-positive bacterial isolated from canine urinary tract infections (UTI) and may be used without prior isolation or enrichment. This platform provides an adaptable and efficient diagnostic tool for antibiotic susceptibility testing.

[1] Department of Food Science, College of Agricultural and Life Sciences, Cornell University, Stocking Hall, Ithaca, NY 14853, USA. [2] Department of Clinical Sciences, College of Veterinary Medicine, Cornell University, 602 Tower Rd., Ithaca, NY 14853, USA. ✉email: Alireza@cornell.edu

Antibiotic resistance (AR) is a rising crisis worldwide. The rise of AR has largely been attributed to the overuse of antibiotics in human and veterinary medicine, and animal husbandry[1,2]. AR can be caused by a number of factors including over-prescribing of antibiotics, a lack of rapid laboratory tests to help identify antibiotic susceptibility, and prescribing either an ineffective or only marginally effective antibiotic[3]. Therefore, advancing the development and use of rapid diagnostic tests to identify and characterize resistant bacteria, leading to targeted antibiotic administration is a key in combating antibiotic resistance.

Currently, the workflow of a bacterial infection diagnosis takes 2–3 days and includes three steps: isolation, identification, and sensitivity[4–6]. It starts with sample collection, and the sample is either processed on location or shipped to an accredited diagnostic lab for sensitivity test. Upon arrival, the sample is plated onto non-selective and/or selective agar to enrich or isolate the bacteria in the sample. Then the infective agent is identified by morphological, biochemical, or molecular assays. This process can take up to 2 days depending on the type of sample, bacteria, and techniques used at each clinical microbiology lab. Then, after the bacteria are isolated, antibiotic susceptibility testing (AST) is conducted, which takes another 16-20 h. The following two phenotypic cultures are the current standard methods for AST: agar disk diffusion methods[7,8], which provide susceptibility results based on clearing-zone diameter, and Broth microdilution (BMD) methods[9] which provide results based on culture turbidity or metabolic activity. Interpretive standard breakpoints are used to evaluate the growth of bacteria in the presence of antibiotics and to determine the pathogenic resistance. Because of the long time from collection of samples to receipt of the results of the culturing, there is often a lack of timely information about antibacterial resistance at the decision making stage of veterinary treatments, especially in emergency cases[10–12]. Physicians and veterinarians treat based on clinical presentation of the infection using antibiotics with a broad spectrum of activity rather than wait for AST results to choose an antibiotic specific to the infection[13,14]. Thus, there is a need to develop new platforms for rapid, inexpensive, and easily implemented AST approaches to reduce the assay time without compromising accuracy so that more targeted antibiotics can be used.

Recently, microfluidic platforms have been shown to improve the sensitivity and speed of AST. These platforms use techniques such as single cell confinement[15–21], microchamber arrays[14,22–28], droplet microfluidics[29–31], and asynchronous magnetic bead rotation[32] to immobilize or contain the bacteria and conduct AST. Microchamber-based platforms allow integration of microfluidic concentration gradient generators, thus offering high-throughput screening of multiple antibiotic concentrations and experimental controls on the same chip. Previous platforms using simple diffusion have produced continuous concentration gradients of antibiotics[33,34], but these approaches present a problem in determining the minimum inhibitory concentration (MIC) precisely. More recent platforms were successful at generating concentration gradients that are linear[33–37], logarithmic[35], or sigmoidal[38]. Nevertheless, these techniques employ unstandardized concentration gradients. They deviate from the recommended standard quantitative AST technique that is based on 2-fold serial dilutions of an antibiotic to produce an exponential gradient[5]. Note that the susceptibility/resistance determination is based on the MIC interpretive breakpoints; the lack of harmonization between these methods and the current standards may present a challenge in interpreting results and consequently the utilization of these platforms.

We have designed and developed a microfluidic system with an optimized ladder shaped network that combines and distributes culture medium and antibiotic in a 2-fold serial dilution. Based upon our previously established protocols for microscale culture-based AST[14,23–26], antibiotic susceptibility can be determined in <5 h using the ladder system. We evaluated the performance of this platform to perform AST of bacterial isolates as called for by the standard method and explored the feasibility of its use on bacteria retrieved directly from canine urine samples without prior isolation or enrichment. This new platform provides an adaptable diagnostic tool and maintains relevance with current national and international interpretive standards.

## Results

**Description of the system**. The ladder microfluidic system for AST combines our method of nanoliter microchamber-based AST[14,23] with a ladder-shape concentration gradient generator (LCGG). This combination provides a standardized and tunable antibiotic concentration profile for rapid phenotypic AST (Fig. 1a). The platform consists of a patterned PDMS layer bonded to a conventional glass slide, allowing testing of one antibiotic/bacteria combination per device. Each device provides ten 2-fold diluted concentrations for testing in triplicate microchambers for each concentration including positive growth control (no antibiotic) microchambers (Fig. 1ai). The microchambers branch from the main channels of the LCGG and serve as bioreactors to incubate bacteria with antibiotics.

The device has three openings: a drug inlet (culture medium containing antibiotic), a negative inlet (pure culture medium), and an outlet (detailed dimensions in Fig. S1). The drug and negative channels are connected by side channels, creating a ladder-like structure, the LGCC. The LGCC controls the convection of fluid to generate a target concentration profile without using external valves; instead connecting loops are used where each loop results in one 2-fold dilution. The antibiotic and culture medium are added to the device from opposite directions, and portions of the negative solution flow to the drug side through the side channels and dilute the incoming drug solution at each node. The antibiotic is sequentially diluted as it moves through the system (from drug inlet to negative inlet). To ensure that the diluting antibiotic solution is homogeneous before exposing it to the bacteria, we used a modified serpentine (Fig. 1aii), to mix the drug and culture media in a specific range of flow rates. An on-chip water bath feature surrounds the entire main device, allowing water vapor exchange through the porous PDMS wall and reducing reagent evaporation during incubation.

**Design and simulation of the device**. A critical engineering challenge of the device was matching the flow rate of the diluter (pure culture media) to the incoming antibiotic stream and to create the 2-fold dilution at each loop. Without any control feature, the diluter would reach only half of the loops as the path of least resistance is a close-by outlet where the fluid can flow out of the system. We solved this by increasing the hydraulic resistances on the side channels, thus driving the diluter to reach all the loops. The approximate value of the resistance which results in desired concentration profile in our designed ladder layout can be calculated using circuit logic modeling[14,39]. Initially the resistance of the main channels was ignored due to their larger dimensions with respect to the rest of the chip features. A more detailed model was later used to evaluate the performance of the design, which included the resistance of the main channels.

At the side channels the flow rates were designed to be a half that of the drug inlet flow rate. For simplicity we refer to flow rates as their ratio to the drug inlet. At each loop, a relative flow rate, 0.5 of the drug, is mixed with a flow rate of 0.5 of the diluent coming from the top main channel. After this mixture passes through the

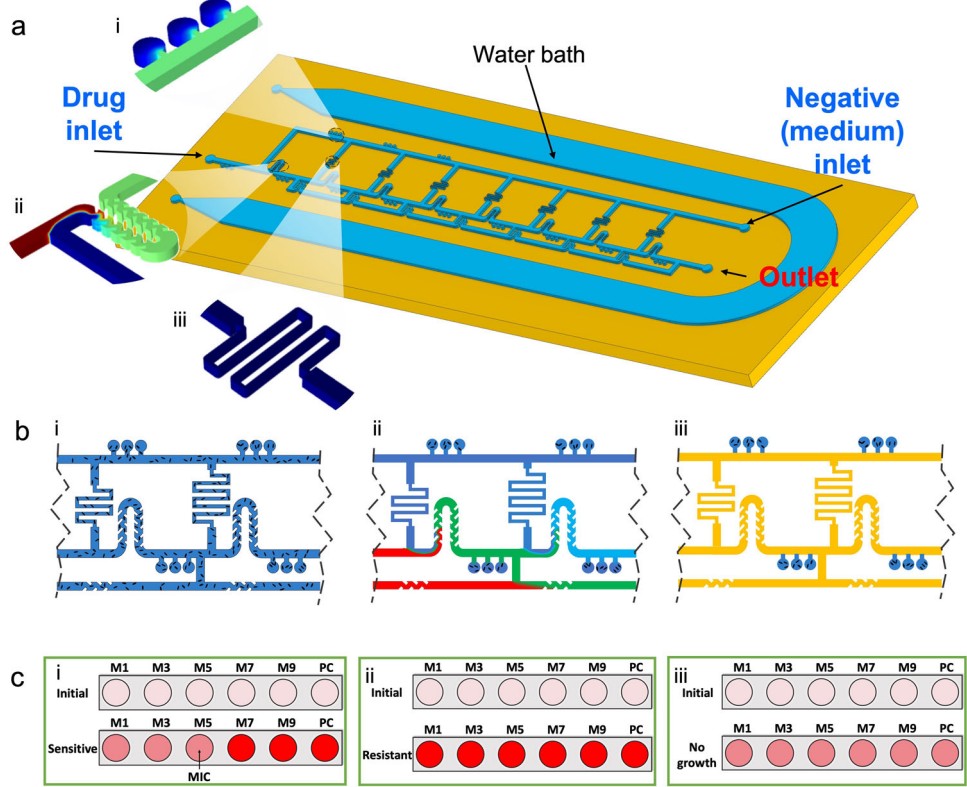

**Fig. 1 Device design and principle of operation. a** A schematic representation of the ladder microfluidic system including an on-chip water bath. Insets showing the key features of the device: (i) the microchamber triplicates, (ii) the modified serpentine mixer, and (iii) the hydraulic resistor pinch on the side channels. **b** The protocol for platform operation: (i) device is loaded with bacteria suspension (blue), then (ii) loaded with culture medium containing antibiotic (red) and culture medium alone (blue) from two different inlets. The LCGG automatically generates a stable exponential decay concentration gradient of antibiotic, which diffuses into the microchambers. Finally, (iii) the channels are washed with oil (yellow), after a certain loading time, to isolate the microchambers. **c** The fluorescent intensities of resazurin in selected microchambers (M1-M9; PC = positive growth control) changes after 4–5 h of incubation at 37 °C indicating bacteria metabolism. Three scenarios are possible where the test shows (i) a minimum inhibitory concentration (MIC) for the bacteria/antibiotic combination, (ii) resistance or MIC higher than the testing range, and (iii) invalid result due to no growth of bacteria.

mixer it flows into a series of 3 microchambers which are attached to the main channel. Right after these microchambers, 0.5 of this flow rate will be removed through a small added section.

To find the resistance and flow within the mixer $R_M$ (Fig. 2a):

$$R_{S_{i+1}} q_s = R_{S_i} q_s + R_M Q_M \tag{1}$$

Where R is the resistance, M is the mixer, q is the flow rate, S is the side channels and $i$ is the number of the loop. Then for each of the loops:

$$R_M q_M = R_{d_i} q_{d_i} \tag{2}$$

where $q_M = 1$; $q_{d_i} = 1 + 0.5i$; $q_s = 0.5$; and d is referred to as the resistances of the added sections. Substitution of the flow rates into Eqs. 1 and 2 provides:

$$R_{S_{i+1}} = R_{S_i} + 2R_M \tag{3}$$

Such that $R_{d_i}(1 + 0.5i) = R_M$ and therefore $R_{d_i} = \frac{2R_M}{2+i}$.

Once the required resistances are determined, the length and width of each side channel is calculated to provide the desired resistance. A constriction (Fig. 1aiii) in the width of a portion of the side channel increases hydraulic resistance which helps streamline the design compared to increasing the length of the serpentine.

We used computational fluid dynamics (CFD) simulations to determine the hydraulic resistance of the constriction as a function of its length (Fig. 2b–d). The CFD simulations showed that a constriction of 40 µm x 655 µm (width x length) has the same hydraulic resistance as a mixer (Fig. 2d). The details of the constrictions' length are depicted in Fig. S1.

After we determined the network layout and the dimensions of all resistors, we evaluated the concentration profile resulting from the network by considering all the features in the chip. Since there are multiple features in the chip, we wrote a generic Matlab code that takes the pressure nodes (the black dots in Fig. 2a), the dimensions of the features that connect them, and the boundary conditions, then calculates the resulting flow rates in each feature and the concentrations of the drug at each node. The code is now available on Github[40]. Using this code, we calculated the concentration profile at various flow rate ratios between drug and diluent (Fig. 2e). We then fitted an exponential equation to the data; concentration (%) $= 50 \times E^{1-i}$, to calculate E as the base of the power. Here $i$ is the number of the loop (1–9). For 2-fold dilution, E needs to be 2 and the model suggests that the flow rate ratio of both inlets should be 7.8 (Fig. 2f).

The ladder microfluidic system is an improvement over our previous designs[14] as it allows us to generate a broader range of concentrations with dilutions over two orders of magnitude. Further, the standardized concentration profile (2-fold dilution) ensures relevance and translatability of the results when interpreted with standard breakpoints set by national and international organizations such as the Clinical and Laboratory Standards Institute or the FDA.

**Principle of operation.** The operation of the ladder microfluidic system follows our previously established protocol.[14,23–26] We

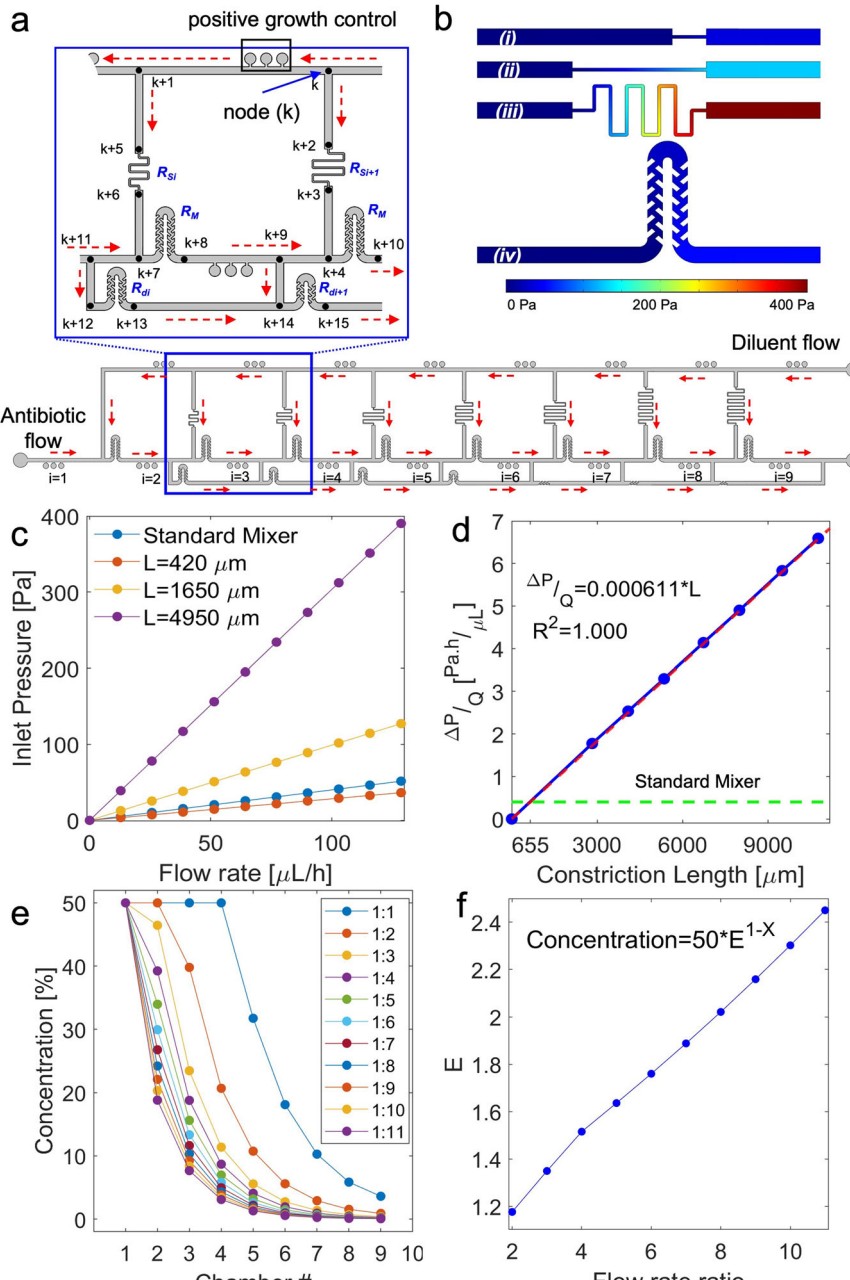

**Fig. 2 Computational fluid dynamics simulation and the pressure nodes network. a** Flow distribution in the ladder. In the inset unique numbers are assigned to the pressure nodes before and after each resistor feature, and at the junctions. The red arrows show the direction of the flow. i and k are counters for the resistors and nodes, respectively. **b** Contours of pressure for constriction of (i) $L = 420\,\mu m$ (ii) $L = 1650\,\mu m$ (iii) $L = 4950\,\mu m$ and (iv) the standard mixer resistors for the flow rate of $130\,\mu L\,h^{-1}$ for each of the channels. **c** The pressure at the inlet of the features in A. **d** Hydraulic resistance of the constriction, as a function of the length for width of $40\,\mu m$. Simulations for seven constrictions are shown ($L = 420\,\mu m$, $1649\,\mu m$, $4070\,\mu m$, $4949\,\mu m$, $7990\,\mu m$, $10795\,\mu m$). At $L = 655\,\mu m$, the hydraulic resistance of the constriction is equal to that of the mixer. **e** Ratio of the flow rates of the drug and the diluent at their respective inlets will affect the concentration profile of the microchambers. These results are from network simulations. **f** The dilution factor in the Concentration$(\%) = 50 \times E^{1-i}$ needs to be $E = 2$, where i is the chamber number. The formula is fitted to the results in the part E to solve for E using least square algorithm.

used the standardized cation-adjusted Mueller Hinton broth to prepare all the solutions, with 2 v/v % PrestoBlue added as a cell metabolism indicator. Loading of the platform consists of the following three steps: (1) a bacterial suspension was loaded into the device using a syringe (Fig. 1bi); (2) an antibiotic solution in culture media and clean culture medium were loaded into the device through two different ports using syringe pumps to generate a concentration profile in the main channel network. This action washes away the bacteria in the main channel but leaves those inside the dead-end microchambers (Fig. 1bii). The antibiotic diffuses into the microchambers according to a drug specific loading time, after which, (3) the system is loaded with biological grade mineral oil to isolate the microchambers (Fig. 1biii). Once the microchambers are sealed off, the on-chip water bath is loaded with water, then the device is incubated at 37 °C for 4–5 h (Fig. S2).

We selected four bacterial species representing pathogens relevant to canine UTI: *Escherichia coli* (EC; $n = 14$), *Proteus*

*mirabilis* (PRM; $n = 8$), *Staphylococcus pseudintermedius* (SP; $n = 5$), *Enterococcus faecalis* (EF; $n = 3$). We prepared the suspensions of these bacteria to be used in the platform at ~$3 \times 10^6$ CFU mL$^{-1}$, which is slightly higher than the standard method which calls for concentrations around $5 \times 10^5$ to $1 \times 10^6$ CFU mL$^{-1}$. We found that this concentration improved sample dispersions in all the microchambers without changing the resulting MICs. We followed the result interpretation protocol as previously established based on the difference in fluorescent intensities of resazurin as a cell metabolic indicator[14,23–26]. We hypothesized that there are three patterns of bacterial growth in the ladder microfluidic system: finding the minimum inhibitory concentration (MIC) (Fig. 1ci), determining antibiotic resistance (or the MIC is higher than the maximum tested concentration, Fig. 1cii), or observing no growth due to a low number of bacteria or an invalid test (Fig. 1ciii). The latter two scenarios can be assessed because we included a positive growth control in the system to validate the results.

During the incubation period, we observed that the background fluorescence of the resazurin increased, however, the fluorescent intensity indicating bacterial metabolism remained distinguishable. Due to differences in growth rate and doubling times between the four species, the signal intensities were different among the Gram-negative and Gram-positive samples. More specifically, Gram-negative bacteria EC and PRM doubled much faster than the other Gram-positive bacteria and therefore had higher overall fluorescent intensities. However, we observed that after the 4–5 h incubation period, the changes in fluorescent intensities were sufficient to determine the MICs for all species. For most of the samples, especially the EC and PRM samples, results could be read after 4 h and as early as 3.5 h. For some of the Gram-positive samples, however, the signal took longer to develop due to slow growth rate, and was given an additional 30 min to 1 h of incubation time. Prolonged incubation, up to 5 h, of the Gram-negative bacteria did not change the MICs.

Among the three steps, bacterial loading, antibiotic loading and microchamber sealing, only the loading time in step 2 (Fig. 3a) changes depending on which antibiotic is being tested. Characterizing the loading of antibiotics into the microchambers and resulting concentration profile were most important in determining the operational procedure of this system. First, we examined the diffusion kinetics of resazurin into the microchambers (Fig. S3). The resazurin molecules diffuse into the side channel ($x_1 = 0–0.2$) then into the main body of the microchamber ($x_2 = 0.2–1$). The concentration of resazurin increased in both regions. At $t = 3$ min, the average concentration inside the microchamber (average area under the curve of $x_2$) was roughly 50% of the $C_0$ in the main channel and was assigned as the specific loading time for resazurin (Fig. 3b). We chose the 50% cut-off for ease of antibiotic preparation, and to reduce the required loading time; resazurin did not reach saturation in the microchamber for almost 7 min. The resazurin concentrations in the ten microchambers form an exponential gradient (Fig. 3c). The concentration at microchamber 2 has the highest error, which corresponds with the oscillations we occasionally observe at the first node of some devices due to differences in the fabrication process. The oscillation, and its standard deviation, however, was reduced as the flow regime became more stable in subsequent loops. Loading time is dependent on the molecular size, therefore, we choose three dyes with different sizes to estimate how antibiotics with comparable sizes will diffuse. We previously demonstrated a linear relationship between loading time ($T$) and molar volume ($V$) such that ($T_1 * V_1 = T_2 * V_2$)[24,26]. Using this, we estimated the loading time for other fluorescent dyes, fluorescein ($208 \pm 4$ cm$^3$ mole$^{-1}$) and Calcein ($356 \pm 5$ cm$^3$ mole$^{-1}$), based on the loading time of resazurin ($145 \pm 7$ cm$^3$ mole$^{-1}$) and measured the concentration

profiles of each molecule under the respective loading time. A linear correlation between the concentration profiles of these three molecules and the theoretical 2-fold dilution concentration profile showed high accuracy with <5% error (Fig. 3d, Table S1). Individual concentrations of these profiles also follow the same trend (Fig. 3e). Based on this operating principle, the loading time for most relevant antibiotics would be equal or <10 min for each. Specifically, the loading times for the antibiotics used in the ASTs in this work ranged from 4 min 47 s to 5 min 57 s. However, the specific loading times only apply if a single antibiotic is used. When drug combinations that contain molecules with different loading rates, such as Amoxicillin/Clavulanic acid and Trimethoprim/Sulfamethoxazole are used, a mismatch in concentration gradients will result if only one of the two antibiotic loading times is considered. In this case, we loaded the solutions of antibiotic mixtures for 10 min to ensure saturation of all molecules in the microchambers.

**Performing AST with clinical isolates**. The performance characteristics of the ladder microfluidic system in determining MICs were examined with bacterial isolates from canine urinary tract infection samples which were submitted to the Cornell Animal Health Diagnostic Center (AHDC) for AST (Fig. 4a). We focused on the four most prevalent infectious organisms in canine UTI cultured at the facility between 2007–2017: *Escherichia coli* (EC), *Proteus mirabilis* (PRM), *Enterococcus faecalis* (EF), and *Staphylococcus pseudintermedius* (SP). These four bacterial species accounted for 66.7% of all cases. Thirty (30) bacterial isolates were tested in total. The representations of the four bacteria in the sample reflects their clinical distributions: EC ($n = 14$), PRM ($n = 8$), SP ($n = 5$), and EF ($n = 3$). We carried out AST of these organisms against the seven antibiotics listed on the Sensititre Veterinary UTI plate (SVU) which is used by the AHDC to analyze these samples: Enrofloxacin, Ceftiofur, Tetracycline, Cefalexin, Ampicillin, and two antibiotic combinations Amoxicillin/Clavulanic acid, and Trimethoprim/Sulfamethoxazole (Table S1). For the purposes of result comparison, the range of concentration and the combination ratios for these antibiotics follow those on the SVU plate. To compare the MICs obtained by AST using the ladder microfluidic system and using the SVU, we followed FDA guidance for characterizing the performance of new AST systems and interpreted the data based on the number of agreements. First, we paired the on-chip MIC and the MIC obtained from the SVU plate as carried out by the AHDC for each bacteria/antibiotic combination. An agreement designation (match) was assigned if the on-chip MIC was within ± one 2-fold dilution. For many instances, the MIC determined by the SVU was in the form of a range, for example, many of the MICs for AST with Trimethoprim/Sulfamethoxazole were ≤ 2 µg mL$^{-1}$. For these pairs, a match was confirmed if the on-chip MIC fulfilled the condition of the SVU plate MIC (for example, on-chip MIC = 0.5 µg/mL was a match for SVU MIC ≤ 2 µg mL$^{-1}$). There were four occasions where we did not have data from the SVU (three for Cephalexin and one for Ceftiofur), so these were excluded from the statistical evaluation.

In total, we evaluated 206 bacterial samples for MIC on both the ladder microfluidic system and the gold standard SVU plates. We observed an overall 91.75% accuracy of determining the MIC between the two methods; 189 samples matched between the two platforms. Further, we calculated the matching rate for each individual bacteria strain as the number of samples that matched (m) divided by the total number of bacteria samples of that type (n). Among the four bacteria, EC and PRM had the highest matching rate at 92.71% (89/96) and 94.54% (52/55), respectively,

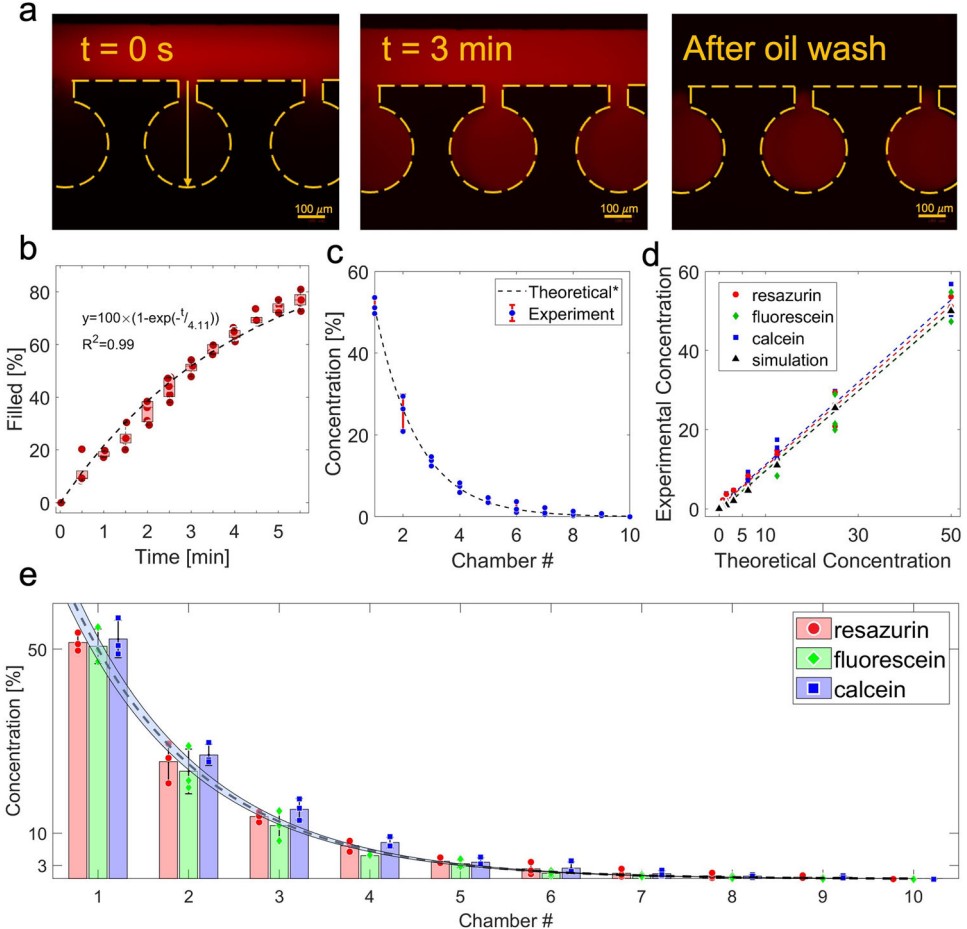

**Fig. 3 Characterization of the concentration gradient. a** images of the microchambers showing the process of molecules diffusing into the microchambers from the main channel at $t = 0$ s and $t = 3$ min, and after the main channel is washed with oil. **b** the average concentration of resazurin diffused into the microchambers over time, represented as the area under the curve of the concentration profile at each time point ($n = 5$). **c** the concentration profile formed in ten microchambers of the ladder microfluidic system after a loading time of three min ($n = 3$). **d** linear correlation between the concentration profiles from computer simulation, of resazurin, fluorescein, and Calcein formed in the ladder microfluidic system, and theoretical 2-fold dilution concentration profile. **e** Comparison of the concentration profiles of resazurin, fluorescein, and Calcein loaded with their specific loading time and theoretical profile. Dash line with blue zone shows the theoretical concentration profile with 5% error. All data are shown as average ±standard deviation.

and SP and EF had the lowest matching rate at 88.57% (31/35) and 85.00% (17/20), respectively.

We also examined the data from the perspective of individual bacterial species, antibiotics, and their combinations (Fig. 4b). When examining the accuracy with respect to each antibiotic, we found that on-chip AST with Ampicillin had the highest MIC accuracy at 100.0%, followed by Cefalexin (96.30%), Ceftiofur (96.55%), Trimethoprim/Sulfamethoxazole (96.67%), and Amoxicillin/clavulanic acid (90.00%). The worst performing antibiotics in terms of aligning with the results from SVU were Enrofloxacin (83.33%) and Tetracycline (80.00%). In the analysis for the probability of correctly obtaining a MIC for each bacteria/antibiotic combination, we saw a similar pattern where EC and PRM had the highest probability, as high as 97% for EC (Fig. 4c). The lowest probability was observed with the EF/Enrofloxacin combination, which is likely inconclusive due to sample size ($n = 3$). Overall, our analysis did not find a statistically significant difference in the probability of achieving an accurate MIC using the ladder microfluidic method in comparison to the SVU.

To further understand the potential sources of errors leading to disagreements in some of the bacteria/antibiotic groups, we repeated those samples which had more than three disagreements. While we saw improvement in accuracy with some samples after repeating, suggesting random handling error in the

first experiment, most of them had the same MICs. The same cannot be said for the original SVU data we obtained from AHDC, which was conducted once. This observation leads us to believe that our method is reliable and repeatable, and the mismatch between the MICs could be due to errors from the SVU as run by the AHDC.

**Direct-from-sample AST with spiked and clinical samples.** While we were able to reduce the time-to-result of the AST of clinical isolates, we argue that for some urine samples, the overall time required for obtaining a susceptibility result can further be reduced, up to 24 h, by bypassing the bacteria isolation step[17,21,38,41–43]. Thus, we evaluated the ability of our device to conduct AST directly with urine samples. The study was conducted with seven urine samples that were cultured positive for the presence of a single unknown bacteria. Because these are canine urine samples, the volumes were limited, ranging from 2-3 mL. Each urine sample (2 mL) was first centrifuged at 13,000 $g$ for 1 min, and the resulting pellet resuspended in 1 mL of fresh MHB. Afterward, the suspension was filtered through a 5 µm syringe filter to remove any debris or eukaryotic cells. We found that this sequence of bacterial collection was the most consistently effective for the limited volume of our samples. The first step

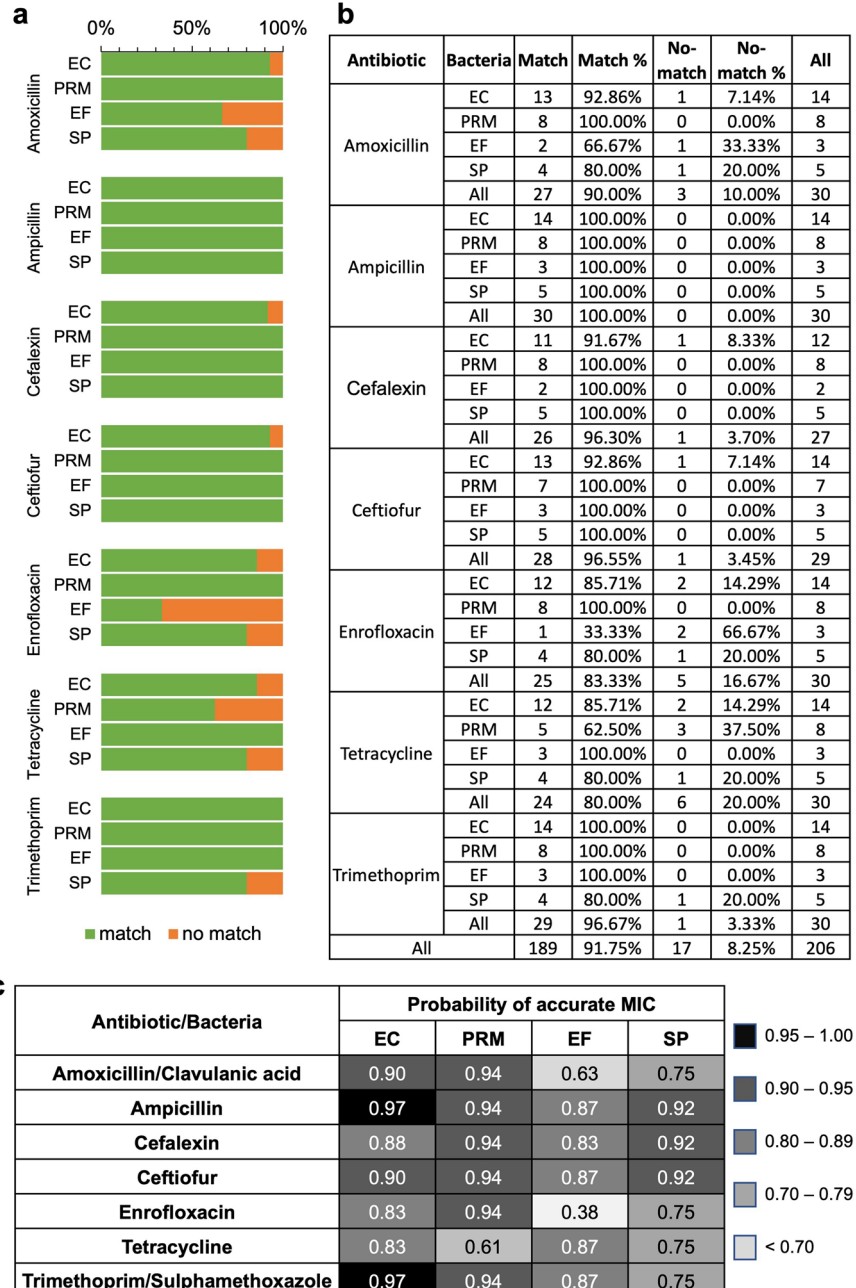

**Fig. 4 Efficacy of microfluidic platform for AST of bacterial isolates from clinical veterinary samples. a** Comparison of match vs. un-match between MICs obtained on-chip using the ladder microfluidic system and the gold standard method conducted at veterinary diagnostic lab. Targeted bacteria are *Escherichia coli* (EC), *Proteus mirabilis* (PRM), *Enterococcus faecalis* (EF), and *Staphylococcus pseudintermedius* (SP). **b** Detailed percentage of matched vs. unmatched for each antibiotic/bacteria combination. **c** Probability of obtaining an accurate MIC for each antibiotic/bacteria combination.

collects and concentrates all the bacteria and cells or debris inside the sample, while the second step isolates the targeted bacteria. This centrifuge-filter sequence leads to a decrease in bacteria loss as the larger pellet from the first spinning step helps in recovering bacterial cells. We found that samples containing > $10^5$ CFU mL$^{-1}$ gave consistent results, and thus excluded the diagnosis low-end of $10^4$–$10^5$ CFU mL$^{-1}$. This limit is considered as the limit of detection of the method when testing directly from urine, however, most of the samples received at the facility are above $10^5$ CFU mL$^{-1}$. Finally, the isolated bacteria were preincubated for 2 h to bring the cells out of lag phase before conducting AST. Similar to the original method with bacterial isolates, we were able to determine the MICs of these bacteria from clinical samples in 4-5 h. We believe that the preincubation

period is necessary when testing bacteria directly isolated from stored patient urine samples, such as the samples in this study, which were refrigerated around 24-36 h before we received them for testing. This period appears to bring the bacteria out of the lag phase induced by refrigeration.

We also tested some culture negative samples and confirmed the lack of false positives by observing no growth-indicative changes in the fluorescent intensity in the positive growth chambers which did not get exposed to antibiotics. Additionally, seven clinical isolates from the previously analyzed samples were spiked into pooled canine culture negative urine, and the same isolation/pre-enrichment process was carried out before performing AST. We observed the same overall rate of agreement between the clinical and spiked samples (91.84%) (Table 1), which is very

similar to the rate observed in our study with the clinical isolates (91.75%) (Fig. 4b). This allowed us to conclude that the ladder microfluidic platform can perform AST for either bacterial isolates or bacteria directly collected from urine samples. In this study, we did not observe a consistent discrepancy between our MICs and those obtained using the SVU plate as performed in the clinic, except for spiked sample 11 which had three discrepancies. Among those three discrepancies, only one (versus cephalexin) was consistent with the result from the first bacterial isolates analysis, leading us to believe that the other discrepancies were most likely the result of random errors such as operation or differences in device fabrication.

When selecting the culture positive urine samples, we did not exclude any sampling method that would generate a liquid volume sample (swab samples were excluded), or type of collection tube, some of which contain additives to stabilize bacterial count. Comparing the results from all the urine samples suggests that the sample collection method does not have a effect on the performance of the method in determining MIC. These results collectively support direct-from-sample AST using the ladder microfluidic system.

## Discussion

We designed, optimized, and used a microfluidic system that performs phenotypic AST; incorporates standardized, translatable testing parameters; and tests both isolated bacteria from culture plates or bacteria isolated directly from urine samples. The system utilizes a ladder shaped network to generate a serially 2-fold diluted concentration gradient that follows current standardized methods and regulations and is tunable to accommodate a broad number of clinically relevant antibiotics and antibiotic combinations. The total preparation time, which includes bacteria loading, antibiotic loading, and oil loading, can be as fast as 10 min when testing with a single antibiotic and no >15 min for an antibiotic combination. An important consideration of the ladder microfluidic system is the two-fold serial dilution of the antibiotics on the microscale and within a small space. Although exponential concentration gradients have been generated in microscale, by adapting traditional gradient generators such as the Christmas tree[44], or H-filter[45], these designs tend to be larger in size. Controlling the fluidic flow in a ladder-shape grid results in a simpler and smaller device, which is also easier to handle. This ladder microfluidic system is the first, to date, to use circuit logic to generate a concentration gradient in microfluidics.

Rapid AST can be performed on both bacterial isolates which is the standard type of sample for AST, or on clinical urine samples that contain the disease-causing pathogen. Compared with standard AST which requires 18–24 h, the time-to-result using this microfluidic platform is shorter; results can be obtained within a single working day. We have shown that the method is >90% effective at determining the MIC of the most common UTI pathogens in canines against disease relevant antibiotics, when examined with 30 bacterial isolates from clinical samples. Specifically, the method was able to determine the MICs of the Gram-negative bacteria E. coli and P. mirabilis accurately up to 92.71% and 94.54%, respectively. The rate of accuracy for the other two Gram-positive bacteria, S. pseudintermedius and E. faecalis that we tested were slightly lower, at 88.57% and 85.00%, respectively, but no statistical significance was found. Further, the method is capable of testing bacteria recovered directly from clinical samples, eliminating the need to culture the pathogen for 12-48 h. Because of the small dimensions at the microscale, our method does require a higher inoculum than the standardized method to minimize sampling effects. This requirement is not an issue when testing bacterial isolates plated on culture plates, but poses a

**Table 1 Efficacy of the ladder microfluidic platform for rapid AST of bacteria extracted directly from spiked and clinical canine urine samples.**

| Sample # | type | Species | Enrofloxacin Chip | Enrofloxacin SVU | Ceftiofur Chip | Ceftiofur SVU | Tetracycline Chip | Tetracycline SVU | Cefalexin Chip | Cefalexin SVU | Ampicillin Chip | Ampicillin SVU | Amoxicillin/Clavulanic acid Chip | Amoxicillin/Clavulanic acid SVU | Trimethoprim/Sulfamethoxazole Chip | Trimethoprim/Sulfamethoxazole SVU |
|---|---|---|---|---|---|---|---|---|---|---|---|---|---|---|---|---|
| 3 | S | Sp | 1 | 1 | 0.5 | ≤0.5 | <0.5 | ≤2 | 0.5 | ≤4 | 0.5 | ≤2 | 2 | ≤2 | 0.25 | ≤2 |
| 6 | S | Ef | 2 | 1 | >4 | >4 | 64 | 64 | 16 | N/A | 1 | ≤2 | 8 | ≤2 | 1 | ≤2 |
| 8 | S | Pm | 0.25 | 0.25 | 0.06 | ≤0.5 | 16 | 16 | 16 | 16 | 0.25 | ≤2 | 2 | ≤2 | 0.125 | ≤2 |
| 11 | S | Sp | >4 | >4 | >4 | >4 | 8 | 64 | >128 | 64 | 32 | 32 | **16** | **≤2** | >8 | >8 |
| 12 | S | Ec | <0.015 | ≤0.03 | 0.5 | ≤0.5 | 1 | ≤2 | 4 | 8 | 2 | ≤2 | 2 | 4 | <0.03 | ≤2 |
| 25 | S | Ec | 0.03 | 0.06 | **1** | ≤0.5 | <0.5 | ≤2 | 8 | 8 | 4 | 4 | 4 | 4 | <0.03 | ≤2 |
| 27 | S | Pm | 0.25 | 0.25 | 0.125 | ≤0.5 | 32 | 64 | 8 | 8 | 0.5 | ≤2 | <0.5 | ≤2 | 0.5 | ≤2 |
| U1 | C | Ec | <0.015 | ≤0.03 | 0.5 | ≤0.5 | 1 | ≤2 | 64 | 128 | **8** | 64 | 32 | =32 | <0.03 | ≤2 |
| U2 | C | Pm | 0.125 | 0.12 | 0.5 | ≤0.5 | 32 | =32 | 16 | 16 | 2 | ≤2 | 2 | ≤2 | <0.03 | ≤2 |
| U3 | C | Sp | 0.125 | 0.12 | 0.5 | ≤0.5 | 0.25 | ≤2 | **0.5** | 8 | 2 | ≤2 | 1 | ≤2 | 1 | ≤2 |
| U4 | C | Ec | 0.03 | ≤0.03 | 1 | 1 | 0.25 | ≤2 | 8 | 8 | 2 | 4 | 8 | 8 | <0.03 | ≤2 |
| U8 | C | Ec | **<0.015** | **0.12** | 1 | 1 | **4** | **≤2** | 8 | 8 | 4 | 4 | 8 | 4 | <0.03 | 4 |
| U9 | C | Ec | <0.015 | ≤0.03 | 0.5 | ≤0.5 | 16 | 32 | 8 | 8 | >256 | >256 | 8 | 8 | >8 | >8 |
| U10 | C | Ec | <0.015 | ≤0.03 | 0.5 | ≤0.5 | 1 | ≤2 | 4 | 8 | 2 | 4 | 8 | 8 | <0.03 | ≤2 |

*EC Escherichia coli, PRM Proteus mirabilis, EF Enterococcus faecalis, SP Staphylococcus pseudintermedius, S spiked sample, C clinical sample, SVU Sensititre Veterinary UTI plate. Values highlighted in bold do not match.

challenge when testing clinical samples especially at the lower UTI diagnostic concentration $10^4$–$10^5$ CFU mL$^{-1}$. Bacteria losses during the recovery process is the biggest contributing factor to this challenge. Therefore, a better recovery method is needed, or testing of isolated and enriched bacteria would be more appropriate for these samples. The method is effective for urine with higher bacteria concentration, which was most of the UTI diagnosed samples.

In a preliminary experiment, we tested with bacterial load from $10^4$ to $10^8$ CFU mL$^{-1}$. We noticed a higher frequency of false negative growth results in the $10^4$–$10^5$ range, most likely due to sampling error because of the small volume of the microchambers. We saw an increase in MIC with the $10^7$–$10^8$ CFU L$^{-1}$ sample because there were too many bacterial cells. Out of the 189 clinical MICs from isolates, 64 of them were discrete values (not a $<=$ or $>$ range). Among them, 19 had higher MIC, 28 had lower MIC, and 17 had the same MIC as the standard method. In total, 9 out of the 64 received the no-match designation. This trend was improved in the later experiments with the direct from urine samples where we had 39 discrete MICs, with 2 higher MICs, 11 lower MICs, and 23 exact MICs, with an overall 5 no-match designations. This improvement is most likely due to better familiarity with the operation of the platform and assay. Interestingly, there was an increase in the frequency of MICs that were lower than the standard method, which is not what we would expect with a higher than standard bacterial load. However, most of these decreases in MIC are within the acceptable range for method comparison. Furthermore, the PDMS material of our device is known to absorb some small molecules, especially hydrophobic molecules[46], however our data neither strongly supported, nor fully rejected, the effect of adsorption on the MICs.

We characterized the performance of the system on urine samples containing a single organism and excluded those with multiple organisms. This is a limitation for many phenotypic AST methods aiming to test directly from samples where the bacteria species is unknown. Depending on the sample collection process, urine samples can be contaminated with commensal bacteria, and therefore it is difficult to confidently identify the presence of the pathogen(s). Further, false classification of antibiotic resistance might occur if one or more of the contaminants is resistant to an antibiotic, resulting in erroneous exclusion of a potentially effective therapy. Conducting AST on multi-organism samples will require more complex result interpretation and may need the judgment of medical professionals for final diagnostic determinations. For these reasons, we did not include multi-organism samples in the scope of this study.

We focused on the four most common pathogens of UTIs at our local veterinary diagnostic center, and the results from these bacteria suggests that the method could be extended to additional species of Gram-negative and Gram-positive UTI pathogens. We demonstrated the usability of this method to test bacterial isolates in place of the current SVU plate, as well as the feasibility to test bacteria retrieved directly from samples at a clinically relevant concentration. UTIs are among the most common bacterial infections dealt with at a clinical microbiology laboratory. While bacteria speciation is required for final susceptibility determination, this method can perform AST without prior information on the bacteria and interpreted after identification of bacteria is confirmed.

A rapid AST method would improve patient outcome and streamline clinical laboratory workflow by 18 h if tested with identified bacterial isolates. Using the rapid urine test, AST can be conducted as soon as a sample is suspected to contain a UTI pathogen, while other identification methods such as MALDI-TOF are carried out simultaneously to identify bacteria, potentially reducing the time to result up to 2 days. Rapid AST methods can be used in place of, or in complement to, the SVU plate

method. A Rapid AST test that could be read within a day would allow veterinarians and medical professionals to provide more targeted initial prescription during early diagnosis. This would reduce the use of broad-spectrum antibiotics and decrease the potential for antibiotic resistance. With improvement in testing time and adherence to industry standards, we believe that this platform can be used at various stages in the clinical workflow, and thus provides another tool for combating antibiotic resistance without compromising patient health or increasing costs.

Medical device approvals require >95% agreement with gold standard devices like the SVU, so while promising, our Ladder chip is still short of acceptable. In the future, automatic loading and sample handling, and real-time image analysis should be incorporated into the ladder microfluidic system to improve the accuracy of the process. In addition, further clinical studies to expand the scope of this platform for AST and direct from sample AST with more relevant bacteria/antibiotics and examining other disease models outside of UTI should be done to explore additional uses of this device in clinical diagnosis.

## Methods

**Clinical samples.** Bacterial isolates (30) from canine urinary tract infections were obtained from the Cornell Animal Health Diagnostic Center (AHDC). The isolates were identified to species level, with antibiotic susceptibilities examined using a Sensititre™ system (SVU) with a urinary plate for isolates of veterinary origins. The 30 isolates include *Escherichia coli* ($n = 14$), *Proteus mirabilis* ($n = 8$), *Staphylococcus pseudintermedius* ($n = 5$), *Enterococcus faecalis* ($n = 3$), and were selected to reflect prevalence of these organisms as observed at Cornell AHDC from 2007–2017 in canine urine. Isolates were subcultured onto fresh blood agar plate prior to testing and kept at refrigeration temperature for no longer than 5 days.

For direct-from-sample testing, de-identified discarded canine urine from AHDC was obtained in March 2022. Samples were refrigerated overnight until confirmed for the presence of bacterial growth by the diagnostic center. Among the samples, four were growth negative and seven were growth positive. The inclusion criteria for growth positive samples were single organism growth with a count of at least $10^5$ CFU mL$^{-1}$. We did not exclude any sample collection method except swabs as this method does not yield fluid samples. The antibiotic susceptibilities of these pathogens bacteria were examined by the AHDC using the Sensititre™ system (SVU) and our microfluidic system, and the results were compared.

**Antibiotics.** We tested seven different antibiotics and antibiotic combinations, following those included in the SVU plate. They were Enrofloxacin, Ceftiofur, Tetracycline, Cefalexin, Ampicillin, and two antibiotic combinations Amoxicillin/Clavulanic acid 2:1 ratio, and Trimethoprim/Sulfamethoxazole 8:152 ratio. Antibiotic solutions were prepared from powdered stocks then aliquoted and stored at −20 °C.

**Microfluidic device fabrication.** The microfluidic device was made from polydimethylsiloxane (PDMS) (Silgard 184; Dow Corning) patterned with standard soft lithography technique and permanently bonded with a glass substrate[47]. SU-8 2075 was used to make the silicon mold. PDMS was mixed at 1:10 ratio of curing agent to polymer and cured at 70 °C for 2 h. We treated the PDMS and glass substrate with plasma for 1 min to make a permanent bond, and then the device was incubated at 70 °C for at least another 30 min to promote the bonding.

**Characterization of the device.** To characterize the diffusion of molecules into the microchambers, we used resazurin. Fluorescent images were taken using a fluorescent microscope (ZOE™ Fluorescent Cell Imager system; Bio-rad, Hercules, California) every 30 sec up to 5 min, then analyzed using ImageJ. To measure the concentration gradient experimentally, we loaded resazurin solutions for a resazurin-specific loading time, then washed the main channel with mineral oil to isolate the microchambers. The fluorescent intensities of the resazurin in the microchambers were analyzed using ImageJ software. Because the concentration profile on the device expands across more than two orders of magnitude (100%–0.3%), we used three resazurin solutions (i.e., 10 µg mL$^{-1}$, 1 µg mL$^{-1}$, 0.1 µg mL$^{-1}$) to characterize the concentration gradient in three zones: high (M1-3), medium (M3-6), and low (M6-9). At each microchamber, we determined the relative fluorescent signal compared to the previous microchamber, then calculated the concentration at each. To examine the ability to generate the designed concentration gradient for different molecules, we loaded fluorescein and Calcein using their molecule-specific loading time, then calculated the concentration profiles generated.

**Numerical simulation and network model solver.** Comsol multiphysics 5.4a software was used for Finite Element Analysis of the equivalent constriction length to the hydraulic resistance of the mixer.

**Table 2 Excel spreadsheet layout for determination of pressure, flow rate and concentration.**

| Node type | Boundary conditions | | |
| --- | --- | --- | --- |
| | Pressure | Flow rate | Concentration |
| Inlet nodes | Leave empty | + If incoming<br>− If outgoing | Should be known value |
| Outlet nodes | 0 | Leave empty | Leave empty |
| Internal nodes | Leave empty | Leave empty | Leave empty |

For the detailed characterization of the network, we developed a generalized computational method to calculate the flow rates and concentrations. Our algorithm resembles the Hardy Cross method of momentum distribution through pipe networks[48]. It was unrealistic to hand-calculate the flow rates and concentrations for all 65 nodes on our chip. The code is uploaded on Github[40], and a brief explanation is offered here.

All nodes, including both ends of the features in the chip and the channel junctions, are assigned a unique number between one and the total number of the nodes ($n$). An Excel sheet should be generated with the first column starting node ($i$) and second column ending node ($j$) for each feature. Each row contains width and length of the feature and in the starting node ($i$) and ending node ($j$) the numbers should be placed in increasing order. The subroutine then reads the excel sheet and creates a resistor matrix ($R$) in which the row index would be the number of the starting node and the column index is the ending node and the value is the resistance based on the following formula.

$$R = \frac{12L}{H W^3} \qquad (4)$$

L, W and H are the length, width and height of the feature respectively.

A second sheet in the Excel file was named "PQC" that contains the boundary conditions of pressure, flow rate and concentration (Table 2).

Using these two sheets, a matrix will be created containing C(n,2) +n rows and columns, in which C(n,2) is the binomial coefficient of 2 from n for each combination of (i,j) from 1 to n. This is the coefficient matrix for the flow rates between each two nodes plus the pressure at each node which are solved for. The first C(n,2) elements are related to flow rates between each two nodes (zero if there is no connection between the nodes) and the remaining n would be the pressures. The following equation regarding the balance of pressure for the connection between i and j can be written:

$$P_i - R_{i,j}q_{i,j} = P_j \qquad (5)$$

for all the combinations of i and j or if nothing is known then $P_i - P_j - R_{i,j}q_{i,j} = 0$. If there is an external flow source attached to a node, we can assume that there is a ghost node outside of the network that is connected to the node via a resistor with resistance of 1. Therefore, the right-hand side of the above equation becomes

$$P_i - P_j - R_{i,j}q_{i,j} = -1 \times q_e \qquad (6)$$

in which $q_e$ is the external flow source amount. Another set of n equations is the mass balance at each node.

$\sum_{k=1}^{n} q_k = 0$; because all incoming flows rates to a node are positive and the outgoing flow rates are negative such that at each node the flow rate sums to zero. With these C(n,2)+n equations the set would be closed and can be solved for pressures and flow rates. Once these are calculated, assuming that all the incoming flows mix thoroughly, we can solve for concentrations at each node.

$$C_k = \frac{\sum_j^{q>0} C_j \times q_{j,k}}{\sum_j^{q>0} q_{j,k}}$$

In which $\sum_j^{q>0} q_{j,k}$ means summation over all the neighbor nodes that have incoming flow rates into the node; that is $q_{j,k} > 0$. To understand how the indexes are managed refer to the code itself.

**On-chip AST**. The device loading procedure for this work was similar to the established protocols in our previous work[14,25,26]. In brief, bacterial inoculum was loaded into an empty device from one of the openings (we used the $C_0$ inlet), while the remaining openings were blocked, to remove air bubbles and to completely fill the device specifically in the microchambers. Then, $C_0$ solution containing a maximum concentration of an antibiotic, and culture medium alone were loaded into the device from two different inlets at flow rates 35 μL h$^{-1}$ and 262 μL h$^{-1}$, respectively (flow rate ratio of ~7.5). The two solutions were loaded for a specific time depending on the antibiotic, to make the designed concentration gradient which diffuses into the microchambers. For most of the antibiotics tested, the loading time was under 6 min, however, the loading time for the Amoxicillin/Clavulanic acid and the Trimethoprim/Sulfamethoxazole antibiotic combinations were set at 10 min. A longer loading time for antibiotic combinations ensures that both antibiotics have saturated the microchambers, as each has a different diffusion rate. Finally mineral oil was loaded from the

inlet to wash out the solutions, create an oil/water cap on the microchambers, and prevent contamination. Prior to incubation, water was added to the surrounding water bath through one of its openings, which helps to reduce evaporation of the reaction inside the microchambers by water vapor exchanging through the porous PDMS walls.

All AST testing using our microfluidic system follows the following procedures. First, the bacterial suspension was prepared and calibrated to a McFarland 0.5 standard using a spectrophotometer, which correlates to a concentration of $1.5 \times 10^8$ CFU mL$^{-1}$. For our study using bacterial isolates, the suspension was prepared by resuspending 2–3 colonies in sterile PBS. Then, 30 μL of the bacterial suspension was added to 1 mL of culture medium (final inoculum concentration ~$3 \times 10^6$ CFU mL$^{-1}$). In all AST experiments, Mueller-Hinton broth supplemented with 2 v/v % PrestoBlue Cell Viability Reagent was used as a culture medium. Each $C_0$ antibiotic solution was prepared by diluting frozen stocks in culture medium to a concentration 2 times that of the maximum concentration on the SVU plate, except for Amoxicillin/Clavulanic acid and Trimethoprim/Sulfamethoxazole antibiotic combinations were diluted to the same as the maximum concentrations on the plate. One device was used for each antibiotic/bacteria combination, nine 2-fold diluted antibiotic concentrations were examined, and we included one positive control where bacteria were not exposed to the antibiotic. We loaded the bacteria, generated the antibiotic concentration gradient, and loaded the oil as described above. Extra oil was added to the top of the inlet/outlet to make an oil droplet and help prevent air from entering the system and disrupting the oil/culture medium interface. The devices were placed into a petri dish with enough water to cover the glass substrate and incubated at 37 °C in an incubator. After 4 h, the red fluorescent intensities of the microchambers on each device were examined to determine the MIC using a fluorescent microscope (ZOE™ Fluorescent Cell Imager system; Bio-rad, Hercules, California). For slow growth samples where the MIC was not clear after 4 h incubation, we incubated them for an additional 30 min to 1 h.

**AST on spiked and clinical urine samples**. Spiked urine samples were prepared by spiking overnight MHB-culture of bacteria into pooled negative growth urines at 10$^6$ CFU mL$^{-1}$. The spiked samples were refrigerated for approximately the same duration that the clinical urine samples would be kept prior to AST testing in our laboratory. Seven of the 30 bacterial isolates were randomly chosen for the spiking study. The bacteria from both spiked and positive growth urine samples were retrieved by a two-step centrifugation/filtration method. First, 2 mL samples were centrifuged at 13,000 g for 1 min, and the pellet was resuspended in 1 mL fresh MHB. Then, the solution was filtered through a 5 μm filter to remove debris and non-bacterial cells. Afterward, the bacterial solution was incubated at 37 °C with shaking at 120 rpm for 2 h. Subsequent inoculum preparation and AST were performed as described.

**Statistics and reproducibility**. We used JMP version 16 to fit a generalized linear model to assess the probability of the two methods matching and how that depended on the drug and the bacteria tested. We used a binomial distribution with a logit link, and fixed effects of antibiotic, bacteria, and their interaction. Because certain drug-bacteria combinations had a 100% match rate, we used Firth Bias-Adjusted Estimates. Other experiments on fluorescent dyes kinetic and concentration gradients were conducted with independently prepared replicates. Data are presented as means and standard deviations.

**Reporting summary**. Further information on research design is available in the Nature Portfolio Reporting Summary linked to this article.

## Data availability
Data is available at: https://doi.org/10.5281/zenodo.7665774.

## Code availability
The hydraulic Calculator code is available at https://zenodo.org/record/6989770#.Y_UnxnbMLGc.

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

## Acknowledgements

We thank Dr. Kelley Donaghy for scientific writing and editing support, Drs. Belgin Dogan, and Shiying Zhang for their assistance, Dr. Craig Altier and Ms. Rebecca June Franklin-Guild of the Cornell AHDC for advice and assistance in clinical sample selection, Dr. Erika Mudrak of the Cornell Statistical Consulting Unit for her generosity involving statistical consultation, and the Cornell NanoScale Facility for providing fabrication facilities and resources (NNCI-2025233). A.V.N. was supported in part by the United States Department of Agriculture's National Institute of Food and Agriculture under award number 2019-38420-28975.

## Author contributions

A.V.N.: conceptualization, data curation, formal analysis, investigation, methodology, writing-original draft, writing- review and editing. M.Y.: conceptualization, data curation, formal analysis, writing-original draft, writing- review and editing. M.A.: conceptualization, methodology, writing-original draft, writing- review and editing. M.D.: data curation, formal analysis, writing-original draft, writing- review and editing. K.W.S.: conceptualization, resources, writing- review and editing. A.A.: conceptualization, funding acquisition, project administration, resources, supervision writing- review and editing.

## Competing interests
The authors declare no competing interests.
