## [Peer Review File · Communications Engineering]

Reviewers' comments:

Reviewer #1 (Remarks to the Author):

In this manuscript, Nguyen et al. present a ladder-shaped microfluidic device which generates a 2-fold serial dilution of antibiotics in order to improve antibiotic susceptibility testing assays. This work is based on the authors' previous work (ref 14), with improvements in the design of microfluidic devices, and following current standardized methods and regulations. To optimize the design of the device, CFD simulations are used to determine the hydraulic resistance of microchannels, and the concentration profile is evaluated by a generic Matlab code. The authors test the system by determining MICs with both bacterial isolates from the standard type of sample for AST, or clinical urine samples. The proposed approach requires less time and shows comparable accuracy, which makes it have high potential in practical applications. Overall, the concept is new and the device is useful. I would like to recommend it for publication in Communications Engineering after the following comments are addressed.

1. In the AST testing, the authors used 3×10^6 CFU/ml bacteria, which is a little higher than the standard concentration. Does it have an effect on MICs?
2. The microchambers are sealed by mineral oil, which may have an influence on the growth rate of bacteria or the MIC. Explanations are needed. Can aerobic bacteria be applied to this sealed microfluidic system?

Reviewer #2 (Remarks to the Author):

In this manuscript, Nguyen et al. propose an innovative microfluidic device for rapid antimicrobial susceptibility testing thanks to an integrated microfluidic concentration gradient generator. This aims to answer a major worldwide public health treat related to antibiotic resistance.

The manuscript is well written. Firstly the authors described the design of their microfluidic device in order to generate an exponential gradient of drugs. They designed it using circuit logic modelling; which is similar to R-2R resistor ladder used in electronics. Then they described the AMR experiments and the results.

I would recommend the editors to accept this manuscript for publication after the authors address these comments:

- 1) Page 11: The authors use the acronyms EC for Escherichia Coli and PRM for Proteus Mirabilis that are defined page 14. They should introduce the acronyms on page 10 when introduce the different bacteria species of the study.
- 2) Page 14: Can the authors give a short description of the antibiotics use in the study as a table in the SI with the following information for example (antibiotic generation, antibiotic mechanism, targeted bacteria, route of administration)?
- 3) Page 14: Can the authors give reference for the FDA guidance related to the SVU?
- 4) Page 18: In order to obtain consistent results, the samples must contain above 105CFU/mL. It is the limit of detection of their method. Can the authors comment about this limit?

General comments

PDMS used to fabricate the microfluidic device is well-known to adsorb small molecules on the surfaces especially hydrophobic molecules. How does this adsorption affect the results? Do the authors perform a coating on PDMS to avoid this absorption?

Can the authors explain how they normalize the resazurin fluorescence intensity according to the number of bacteria in the microchamber? Does the microchambers contain the same number of bacteria?

Can the authors add simulation points on the plot Figure 2.D?

Reviewer #3 (Remarks to the Author):

Authors designed a microfluidic ladder based system that generates a twofold serial dilution of antibiotics comparable to current national and international standards, while other microfluidic-based techniques employ unstandardized concentration gradients. The manuscript is well written and presented results are solid. The advantage, the limitation, and future direction of this work are clearly discussed. I recommend publication of this manuscript.

Response to Reviewers

The authors thank the reviewers for their time and careful review of our work. We have used the comments to improve the clarity of our manuscript and have made modifications as recommended by the reviewers. The authors' responses to the comments raised by reviewers are listed below, and changes have been tracked in the manuscript.

Reviewer #1 (Remarks to the Author):

In this manuscript, Nguyen et al. present a ladder-shaped microfluidic device which generates a 2-fold serial dilution of antibiotics in order to improve antibiotic susceptibility testing assays. This work is based on the authors' previous work (ref 14), with improvements in the design of microfluidic devices, and following current standardized methods and regulations. To optimize the design of the device, CFD simulations are used to determine the hydraulic resistance of microchannels, and the concentration profile is evaluated by a generic Matlab code. The authors test the system by determining MICs with both bacterial isolates from the standard type of sample for AST, or clinical urine samples. The proposed approach requires less time and shows comparable accuracy, which makes it have high potential in practical applications. Overall, the concept is new and the device is useful. I would like to recommend it for publication in Communications Engineering after the following comments are addressed.

1. In the AST testing, the authors used 3×10^6 CFU/ml bacteria, which is a little higher than the standard concentration. Does it have an effect on MICs?

We did not notice any difference in the MIC. In a preliminary experiment (data not included), we tested with bacterial load from 10^4 to 10^8 CFU/mL. We noticed a higher frequency of false negative growth results in the 10^4 - 10^5 range, most likely due to sampling error because of the small volume of the microchambers. We saw an increase in MIC with the 10^7 - 10^8 CFU/L sample because there were too many bacterial cells. In fact, out of the 189 clinical MICs from isolates, 64 of them were discrete values (not a \leq or $>$ range). Among them, 19 had higher MIC, 28 had lower MIC, and 17 had the same MIC as the standard method. In total, 9 out of the 64 received the no-match designation. This trend was improved in the later experiments with the direct from urine samples where we had 39 discrete MICs, with 2 higher MICs, 11 lower MICs, and 23 exact MICs, with an overall 5 no-match designations. This improvement is most likely due to better familiarity with the operation of the platform and assay. Interestingly, there was an increase in the frequency of MICs that were lower than the standard method, which is not what we would expect with a higher bacterial load than standard. However, most of these decreases

in MIC are within the acceptable range for method comparison. Further investigation can help us better understand this phenomenon and improve the method.

We added this discussion to the revised manuscript.

2. The microchambers are sealed by mineral oil, which may have an influence on the growth rate of bacteria or the MIC. Explanations are needed. Can aerobic bacteria be applied to this sealed microfluidic system?

The system is aerobic as only the “neck” of the microchamber is sealed by mineral oil. The remaining body of the microchamber is surrounded by PDMS which is gas permeable.

Reviewer #2 (Remarks to the Author):

In this manuscript, Nguyen et al. propose an innovative microfluidic device for rapid antimicrobial susceptibility testing thanks to an integrated microfluidic concentration gradient generator. This aims to answer a major worldwide public health treat related to antibiotic resistance.

The manuscript is well written. Firstly the authors described the design of their microfluidic device in order to generate an exponential gradient of drugs. They designed it using circuit logic modelling; which is similar to R–2R resistor ladder used in electronics. Then they described the AMR experiments and the results.

I would recommend the editors to accept this manuscript for publication after the authors address these comments:

1) Page 11: The authors use the acronyms EC for Escherichia Coli and PRM for Proteus Mirabilis that are defined page 14. They should introduce the acronyms on page 10 when introduce the different bacteria species of the study.

The suggested changes have been made

2) Page 14: Can the authors give a short description of the antibiotics use in the study as a table in the SI with the following information for example (antibiotic generation, antibiotic mechanism, targeted bacteria, route of administration)?

A table has been added to revised supplementary material.

3) Page 14: Can the authors give reference for the FDA guidance related to the SVU?

The FDA does not have any specific guidance on the SVU. The Sensititre system and its plates and components have received approval from the FDA for use in AST. Information about the procedures and set up of the SVU plate can be found in the manufacturer's booklet <https://assets.thermofisher.com/TFS-Assets/MBD/brochures/Sensititre-Plate-Guide-Booklet-EN.pdf>.

4) Page 18: In order to obtain consistent results, the samples must contain above 10⁵CFU/mL. It is the limit of detection of their method. Can the authors comment about this limit?

Yes, this is the limit of detection of the method when testing directly from urine. This limit may exclude some samples, but this exclusion might not be detrimental to the method as the majority of the urine sample received at the facility contained more than 10⁵ CFU/mL of bacteria. We have added this discussion to the revised manuscript.

General comments

PDMS used to fabricate the microfluidic device is well-known to adsorb small molecules on the surfaces especially hydrophobic molecules. How does this adsorption affect the results? Do the authors perform a coating on PDMS to avoid this absorption?

Thank you for the suggestion. The PDMS was not coated. We are aware of this phenomenon, however our data neither strongly supported nor rejected the effect of adsorption on the MIC.

Further study will help us better characterize this impact of small molecule adsorption on our system. We have added this discussion to the revised manuscript.

Can the authors explain how they normalize the resazurin fluorescence intensity according to the number of bacteria in the microchamber? Does the microchambers contain the same number of bacteria?

We did not normalize the resazurin fluorescence intensity according to the number of bacteria. Rather, the increase in the signal after 4-5 h was used as a qualitative indicator of cell growth. In our previous published work, we presented the changes in normalized fluorescence intensity in the microchambers overtime (figure below; Nguyen et al., 2021). In a preliminary experiment (data not included), we tested the platform with bacterial load from 10^4 to 10^8 CFU/mL. We noticed a higher frequency of false negative growth results in the 10^4 - 10^5 range, most likely due to sampling error resulting in some microchamber not loaded with any bacteria or too few numbers. At the bacterial concentration 10^6 CFU/mL, we saw a more homogeneous distribution of the bacteria in each of the microchambers.

Can the authors add simulation points on the plot Figure 2.D?

The modification has been made

Reviewer #3 (Remarks to the Author):

Authors designed a microfluidic ladder based system that generates a twofold serial dilution of antibiotics comparable to current national and international standards, while other microfluidic-based techniques employ unstandardized concentration gradients. The manuscript is well written and presented results are solid. The advantage, the limitation, and future direction of this work are clearly discussed. I recommend publication of this manuscript.

The authors thank the reviewer for their generous recommendation.

REVIEWERS' COMMENTS:

Reviewer #1 (Remarks to the Author):

The authors have addressed all my concerns. Publish as it is.

Reviewer #2 (Remarks to the Author):

The authors answers all the questions raised by the reviewers and consequently modified the manuscript. I am glad to recommend the editors to accept this manuscript for publication.